# Landsat-derived bathymetry of lakes on the Arctic Coastal Plain of Northern Alaska

Claire E. Simpson[1], Christopher D. Arp[2], Yongwei Sheng[1], Mark L. Carroll[3], Benjamin M. Jones[2], Laurence C. Smith[1,4,5]

[1]Department of Geography, University of California, Los Angeles (UCLA), Los Angeles, California, 90095, USA
[2]Water and Environmental Research Center, University of Alaska, Fairbanks, 306 Tanana Loop Rd., Fairbanks, Alaska, 99775, USA
[3]Computational and Information Science and Technology Office, NASA-GSFC, Greenbelt, Maryland, 20771, USA
[4]Department of Earth, Environmental and Planetary Sciences, Brown University, 324 Brook St, Providence, Rhode Island, USA, 02912
[5]Institute at Brown for the Environment and Society, Brown University, 85 Waterman St, Providence, Rhode Island, USA, 02912

*Correspondence to*: Claire E. Simpson (clairesimpson17@gmail.com)

**Abstract.** The Pleistocene Sand Sea on the Arctic Coastal Plain (ACP) of northern Alaska is underlain by an ancient sand dune field, a geological feature that affects regional lake characteristics. Many of these lakes, which cover approximately 20% of the Pleistocene Sand Sea, are relatively deep (up to 25 m). In addition to the natural importance of ACP Sand Sea lakes for water storage, energy balance, and ecological habitat, the need for winter water for industrial development and exploration activities makes lakes in this region a valuable resource. However, ACP Sand Sea lakes have received little prior study. Here, we collect in situ bathymetric data to test 12 model variants for predicting Sand Sea lake depth based on analysis of Landast-8 Operational Land Imager (OLI) images. Lake depth gradients were measured at 17 lakes in mid-summer 2017 using a HumminBird 798ci HD SI Combo automatic sonar system. The field measured data points were compared to Red-Green-Blue (RGB) bands of a Landsat-8 OLI image acquired on 8 August 2016 to select and calibrate the most accurate spectral-depth model for each study lake and map bathymetry. Exponential functions using a simple band ratio (with bands selected based on lake turbidity and bed substrate) yielded the most successful model variants. For each lake, the most accurate model explained 81.8% of the variation in depth, on average. Modeled lake bathymetries were integrated with remotely sensed lake surface area to quantify lake water storage volumes, which ranged from $1.056 \times 10^{-3}$ to $57.416 \times 10^{-3}$ km$^3$. Due to variation in depth maxima, substrate, and turbidity between lakes, a regional model is currently infeasible, rendering necessary the acquisition of additional in situ data with which to develop a regional model solution. Estimating lake water volumes using remote sensing will facilitate better management of expanding development activities and serve as a baseline by which to evaluate future responses to ongoing and rapid climate change in the Arctic. All sonar depth data and modeled lake bathymetry rasters can be freely accessed at https://doi.org/10.18739/A2SN01440 (Simpson and Arp, 2018) and https://doi.org/10.18739/A2HT2GC6G (Simpson, 2019), respectively.

## 1 Introduction

The Arctic Coastal Plain (ACP) of Alaska is distinguished by the presence of thousands of lakes, many of which are the product of thermokarst processes (Hopkins, 1949). Thermokarst is the melting of ice in permafrost, resulting in thaw settlement and land surface subsidence (van Everdingen, 1998); such activity may lead to the development of thermokarst lakes (Hopkins, 1949; Jorgenson and Shur, 2007). While thermokarst lakes on the ACP

typically reach maximum depths between 1 to 3 m (Hinkel et al., 2012), an anomalous group of lakes on the ACP
approach depths up to approximately 25 m.

We collected depth measurements and mapped bathymetry at a group of deep lakes located on the Pleistocene Sand Sea (Fig. 1), a distinctive region of the ACP named for its foundational Pleistocene-aged sand sheet and sand dunes (Carter, 1981; Williams, 1983; Williams et al., 1978). Located west of the Colville River, this region spans approximately 15,000 km$^2$ and contains over 16,000 lakes (Jorgenson et al., 2014). The underlying
dune field impacts the regional lithology and lake morphology. Lakes here are nestled between the crests of sand dunes and display a form distinct from that of lakes across the rest of Alaska's North Slope (Hinkel et al., 2005; Jorgenson and Shur, 2007). Deep central basins and wide, shallow littoral shelves surrounded by bluffs distinguish Sand Sea lakes from lakes that have formed in ice-rich permafrost terrain. Studies by Livingstone (1954), Rex (1961), Carson and Hussey, (1962), and Carson (1968) assert that the bluffs around lakes erode by winds, which
carry sand from the bluff faces into the lakes, forming characteristic sandy littoral shelves. These shelves only reach depths of up to three m, whereas the central basins of such lakes can reach depths over eight times that. Due to this striking depth contrast, the distinction between littoral shelves and central basins is apparent in satellite imagery of most lakes in the area (given low-wind and ice-free conditions). Understanding the geological context and morphology of Sand Sea lakes is important when interpreting their spectral signatures in remotely sensed imagery.
We present a dataset to help fill the gap concerning lake depth - particularly deep lake depth - measurements in Arctic regions. By leveraging the *in situ* dataset to tune linear spectral-depth models at individual lakes, we produce lake-wide bathymetry maps and integrate these modeled depths across each lake to quantify water volumes. Finally, we assess spectral-depth similarity in lakes across the Sand Sea to evaluate the prospects of regional water volume modeling. Bathymetry measurements and associated estimates of water volume such as those
provided in our datasets are important when evaluating aquatic habitats, conducting industrial activities that require local freshwater supplies (i.e. ice road construction), and understanding regional water and energy balance. Compared with lakes in surrounding regions of the ACP, Sand Sea lakes tend to be deeper and thus less likely to freeze to the bottom during the winter. Their notable depth means that Sand Sea lakes tend to have lower evaporative losses and are more likely to have basins characterized by floating (rather than bedfast) ice in the winter
(Arp et al. 2015; Engram et al., 2018). These unfrozen lake basins provide crucial overwintering habitat for fish and other aquatic life (Jones et al., 2009; Sibley et al., 2008). Furthermore, liquid water is essential for industry during winter, primarily for ice road construction, but also for ice airstrip and ice pad construction, exploratory oil-well drilling, and withdrawal of water for drillers' and researchers' in-camp use (Jones et al., 2009). Unfrozen winter lakes can also store more heat, affecting the regional energy balance (Jeffries et al., 1999). Therefore, depth and
volume quantifications of deep Sand Sea lakes can help monitor fish habitat and direct locations of water extraction for wintertime infrastructure and consumption for other purposes.

Previous studies have evaluated water depth and bathymetry of lakes in nearby regions using various methods, but are limited either to shallow lakes or by coarse depth resolution (e.g. Hinkel et al., 2012; Jeffries et al., 1996; Jones et al., 2017; Kozlenko and Jeffries, 2000; Sellman et al., 1975). Such limitations make deep lake depth
and volumetric estimation unfeasible. For example, Jeffries et al. (1996) used satellite imagery and radar data to

determine which lakes in regions near Utqiaġvik (Barrow) and Atqasuk, Alaska (including lakes in this paper's study area) froze to the bottom during the winter, extrapolating from their results a classification of lakes as being less than or greater than 2.2 m deep. When used in concert with an ice-growth model, this provided a proxy for coarse lake volume estimation, but was limited to shallow lakes. Hinkel et al. (2012) measured in situ bathymetry for 28 lakes. However, the maximum lake depth of this study on the inner ACP was 2.3 - 5.2 m. Thus, our dataset is unique in its consideration of deep lakes. Furthermore, while optical remote sensing-based retrieval of bathymetry (applied to create our bathymetry maps) is a well-documented approach (e.g. (Clark et al., 1987; Hodúl et al., 2018; Pacheco et al., 2015; Pope et al., 2016; Yunus et al., 2016), in part due to limited data acquisition, such methods have historically been challenging to apply in our study area. One of the model variants we employ was successfully used to extrapolate bathymetry in tropical and sub-tropical coastal marine environments (Jagalingam et al., 2015; Stumpf et al., 2003), however to our best knowledge, the model has never been applied to high Arctic lakes. Volumetric estimates with the resolution provided here (30 m horizontal; 0.03 m vertical) have never been attempted for Pleistocene Sand Sea lakes and the method of depth derivation used in this paper has not been employed in the Arctic.

## 2 Data and Methods

### 2.1 Depth data acquisition

Depth points were sampled across 19 lakes during a field expedition between 22 July 2017 and 27 July 2017. The method of data collection required landing on each target lake in a float plane. A HumminBird 798ci HD SI Combo automatic sonar unit was attached to the back of a float and sampled depth as the plane taxied or drifted across the lake. Depth points were each measured discretely as part of a depth-gradient transect and were sampled at a frequency of one point per second with an accuracy of 0.03 m (due to intrinsic machine error). The number of points collected per lake is specified in Table 1.

Lakes were targeted that were large enough for a float plane to land on in windy conditions (i.e. > ~1 km$^2$ lake surface area), and showed the presence of a distinct littoral shelf and a deep basin on 2.5-m color-infrared aerial photography (U.S. Geological Survey Digital Orthophoto Quadrangles [DOQs]). A single straight transect line was mapped across each target lake prior to field visits to encompass a wide depth range, however due to windy conditions, such lines were not always followed (Fig. 2). Nevertheless, in all but two lakes, a depth range from the littoral shelf to the deep central basin was captured (Table 1). It should be noted that, as transects were comprised of individual points whose relationship to one another was unimportant to the modeling, the direction, angle, and other qualities of the transect are significantly less important than the range of depths captured.

### 2.2 Depth data processing

Depth data points from 17 of the 19 sampled lakes were compiled into a single file to facilitate initial processing, with the lake IDs maintained in the database for lake-specific analysis. Two lakes at which sampling occurred contained an insufficient number of measurements to justify modeling their bathymetry (models produced for these two lakes would have been strongly overfitted). The dataset was then filtered to 13,735 depth points: for

each transect collected with the HumminBird sonar, discrete points were evaluated relative to the depths of their neighbors and anomalous and zero depth points were manually removed from the dataset. This step mitigated sonar errors and improved the smoothness of the bathymetric profiles that were generated from each transect. Subsequently, depths collected at the margins of two lakes at the Pik Dunes (70.234 °N, 153.183 °W) were removed from the dataset after manual inspection due to their anomalous spectral signatures. The unique, white color of the sandy substrate at this group of lakes and the extreme shallow nature of the littoral shelves (~ 0.5 m deep) produces a spectral signature near the margins of lakes in the Pik Dunes area that is easily confused with that of the surrounding land and thus should not be used to analyze lake depth. These Pik Dunes depth points represent outliers and had they been included, our models would have had to reconcile associating strikingly different spectral values with similar depth values. This likely would have decreased overall model performance with the only potential benefit of modeling a limited number of marginal pixels more accurately.

### 2.3  Landsat image selection

Landsat-8 Operational Land Imager (OLI) imagery was chosen for comparison with measured depth data due to its large swath, 30-m spatial resolution, and quality (as assured by U.S. Geological Survey pre-processing). A cloud-free Landsat image (LC08_L1TP_077011_20160805_20170222_01_TI ) was selected that both covered the study area and was acquired on 5 August 2016, that is, at a similar time of year to that of field data collection from the following year (suitable imagery was not available for 2017). The late summer was chosen to provide data for a time when lakes are at an intermediate level, that is, lakes are free of ice, but have not yet reached their lake level minimums (determined when evaporation exceeds precipitation; Jones et al., 2009). It should be noted that water volume varies seasonally and interannually in accordance with precipitation of the preceding twelve months, and therefore the estimated depth data may not be representative of the lake levels year round or from year-to-year. Nevertheless, these variations in lake level are relatively small, with surface area changes often around 0.6% of total surface area (Jones et al., 2009). Furthermore, of these area changes, the majority of change occurs at the shallow littoral shelves and therefore results in little volume change (Jones et al., 2009).

As no ice-free, cloud-free Landsat images exist that cover all study lakes for late summer 2017, we selected a Landsat image from 2016 in order to maximize the number of lakes included at which field data exists, i.e. the number of lakes at which we could model volume. One potentially promising Landsat image exists that covers our study area, however (a) it was acquired at the end of June, just after ice out, when the lake levels are at a seasonal high, and (b) slight cloudiness over some study lakes produced models that predicted depths up to 48% less accurately. The use of 2016 imagery is further justified as the interannual depth and volume changes are smaller than our error metrics. When considering one representative lake (located at 70.147 °N, 151.765 °W),  a continuous depth logger recorded a depth difference of 0.03 m (or 1% of the annual average depth at that point) between the imagery acquisition date (5 August 2016) and the time of data collection (26 July 2017). This represents a smaller depth difference than the 0.05 m difference measured between 30 June 2017 and 26 July 2017. The maximum observed depth change at this location between 1 January 2016 and 26 July 2017 was on the order of 1 m. Observation of an imagery timeseries of a different group of lakes that are typically highly responsive to water level

changes (located at 70.539 °N, 152.733 °W) similarly revealed lake level conditions to be more comparable between 5 August 2016 and 21–27 July 2017 than between these latter dates and 1 July 2017. Overlaying lake surface area changes on an airborne LiDAR-derived digital surface model showed a change in water level of ~ 0.10 m between 5 August 2016 and 1 July 2017, indicating a depth change well within our error margins (Alaska North Slope LiDAR Data - Project Code ALCC2012-05).


2.4 Landsat image processing and analysis

Study lakes were visually assessed in ArcGIS to provide a Boolean turbidity rating for the purpose of analyzing the success of different models. Lake clarity was determined by comparing the selected Landsat image (as an RGB true color composite) with a Landsat image acquired 13 July 2016 (23 days prior to the acquisition date of

the selected Landsat image), as well as color-infrared aerial photographs (DOQs) with a 2.5-m horizontal resolution (Fig. 3). Lakes that showed presence of sediment plumes or water cloudiness near the site of in situ data collection on the selected Landsat image were designated as turbid. Lakes which displayed minimal suspended sediment distant from the area at which depths were recorded were designated as turbid as well, however they were analyzed as if they were clear, as the impacts of sediment would not be seen in the depth point-derived spectral signatures.

Lakes that did not have sediment plumes were designated as clear.

We validated our qualitative visual turbidity assessment using the ACOLITE (software developed at the Royal Belgian Institute of Natural Sciences for aquatic applications of Landsat and Sentinel-2) implementation of the Total Suspended Matter (TSM) algorithm (Nechad et al., 2010). This algorithm provided quantitative support, agreeing with the visual assessment in 14 out of 17 lakes. However this algorithm proved highly sensitive to depth

(spearman rank order correlation = -0.774; p-value < 0.001) and did not detect sediment in deeper waters to the same extent as shallow waters, effectively ignoring the sediment plumes identified visually. Furthermore, the majority of shallow waters were assigned high TSM values by the algorithm, making the differentiation by turbidity at the lake-wide level irrelevant. Considering the points in our transects, 91% of high-sediment (i.e. TSM values in or above the 75th percentile) points had measured depths < 2 m and only five outlier high-sediment values were

detected in points with depths > 4.6 m. To directly address the sediment content in deeper waters, the mean TSM value was calculated at each lake from sample points with depths > 2m. Seven out of eight lakes with the highest average TSM values had been designated as turbid by our qualitative assessment (note that one of these lakes was designated as turbid away from the sampling site – this is counted as an error). In addition, all but one of the nine lakes with the lowest mean TSM values were designated as clear at the sampling site.

The chosen Landsat image was clipped to the study area and a Normalized Difference Water Index (NDWI) water mask was created using ArcGIS tools to subset our study lakes from the surrounding land pixels (McFeeters, 1996). Each of our study lakes were then extracted to individual geoTIFF files for use in bathymetry map production.

2.5 Spectral-depth point extraction

Top-of-Atmosphere (TOA) reflectance values from the blue band (band 2; 452 - 512 nm), green band (band 3; 533 - 590 nm), and red band (band 4; 636 - 673 nm) of the Landsat image were extracted to each point. Although Surface Reflectance (SR) imagery was available, we elected to use TOA reflectance initially because SR algorithms are often suboptimal when looking at water bodies due to the low level water leaving radiance and furthermore, we

are working at high altitudes, where SR corrections are unreliable. Upon comparison, the SR and TOA reflectance values in our selected RGB imagery (discussed below) were very similar ($R^2 > 0.99$) at our sample locations. The coastal band (band 1; 435 - 451 nm) was not included here as there was no basis for its examination in prior similar studies (e.g. Jagalingam et al., 2015) at the time this analysis was conducted and unexpectedly, preliminary results were not greatly improved by the inclusion of the coastal band.

To minimize error caused by associating a single pixel's spectral signature with multiple depth points (i.e. to reduce compatibility issues between the spatial resolution of the sonar transects and Landsat imagery to which the depth points were compared), the dataset was resampled to include only one depth per pixel. This depth was calculated by averaging the sonar depths of all measurements within the pixel, removing depths greater than one standard deviation from this average, and re-calculating the depth mean of the pixel. Aggregating per-pixel

measurements allowed us to identify the dominant depth represented by the pixel's lake color and improve the precision of training data (i.e. reduce the range of input depths associated with a given band ratio). This pixel-representative depth point provides the final depth value used in analysis. All data visualization and manual data editing was undertaken using ArcMap; automated data editing was done with the aid of ArcGIS and python.

2.6 Model application for lake bathymetry mapping

Twelve variations of a spectral-depth algorithm were examined to model bathymetry, each characterized by a specific band ratio, adjustment factor, and growth factor (Table 2). More specifically, the blue to green, blue to red, and green to red band ratios were considered. Such ratios were either simple (e.g. blue band/green band) or transformed according to Stumpf et al. (2003):

$$\frac{\ln (nR_i)}{\ln (nR_j)} \qquad\qquad (1)$$

where $R_i$ and $R_j$ represent the TOA reflectances for bands i and j, respectively. A constant $n$ is included to effect a positive output (Stumpf et al., 2003). We set $n$ to 500, as it ensured that the logarithm would be positive given any feasible band value input, $R$, from our image.

The band ratio and the depth measurement of the point at which the spectral signature was extracted were

correlated using either a linear regression or an exponential function (Fig. 4a-c). The constants obtained from each of these models became the parameters with which to tune the linear or exponential equations for the validation data. The root mean squared error (RMSE) of each regression between input depths and input band ratios provided error statistics for modeled depths. In summary, the twelve model variations were each characterized by (1) one of three band ratios, (2) one of two transformation methods, and (3) one of two growth relationships (Table 2).

For each lake, half of the depth points were semi-randomly selected as input data while the remaining data were used for validation purposes. To ensure that the model was trained and validated with data spanning the full range of input depths, however, the maximum and minimum depths were assigned to the group of data to input into

the model, while the second deepest and second shallowest depth points were retained in the list of validation data. To obtain the best regional model, this same process was undertaken (i.e. selection of half of the data to train the models; application of each of the 12 models), however a sufficient number of depth points exist in the full dataset such that the explicit assignment of extreme depths values to input and validation data was unnecessary (i.e. the selection was fully random).

Each of the 12 models was tested at each of the 17 lakes and on a regional scale. To account for the slight variations in each model's capacity for depth prediction given different random sets of training data, 1,000 trials were performed. This allowed us to assert that the model designated as the most accurate model for a given lake (as determined from one trial) was the same model that most frequently produced the best results for that lake. The best model for each lake, as evaluated by the coefficient of determination between target and predicted data, was used to calculate depths at each pixel in that lake and produce bathymetry raster maps. Depths were multiplied by 900 m$^2$ (the area of one Landsat pixel) and integrated to quantify the lake's water volume. A summary of the data acquisition, processing,and analysis steps is provided in Figure 5.

The most accurate models (i.e. the models that were best able to determine lake depth for the greatest number of lakes) were models with an exponential growth factor with input band ratios blue/green or blue/red (Table 3). In all but three of the study lakes, an exponential relationship was found between spectral signature and depth. At only two lakes did the green to red band ratio provide the best results. The transform ratio provided the best results in 4 out of the 17 lakes, while the simple ratio was used to best model depths in the remainder of the lakes. The difference between the modeled results of the pure versus transform ratio was marginal however, with an average difference between R$^2$ values generated by the respective models of 0.016.

Unsurprisingly, the blue band proved to be the most useful in determining depth overall, while the red band was useful in the presence of turbidity. The blue band was used to tune depths at all but two lakes. Blue light has a shorter wavelength, and consequential higher energy, which allows it to be absorbed less in water than either green or red light. Thus, the reflectance of the blue band decreases less than either the green or red bands in proportion to increasing depth. In contrast, red light is able to penetrate only several meters into most types of water before it becomes absorbed. The red band proved useful in distinguishing depths at both the sandy littoral shelves, where water is typically 0.5 - 3 m in depth, and where suspended sediment was present in the water. As sand reflects red light more than blue or green light and suspended sediment can reduce penetration of blue or green wavelengths in deeper water, this is expected. All of the eight lakes at which the blue to green band ratio provided the best result were free of sediment where measurements were taken. Furthermore, all of the seven lakes designated as turbid at the data collection site required incorporation of the red band to achieve the best depth prediction. One anomalous lake at which no sediment was detected required incorporation of the red band to predict depth most accurately.

The two lakes at which the green/red band ratio best tuned the model were unique in terms of physical factors or sampling locations. One of these lakes showed the presence of an unusual purple-red patch on a shelf between the littoral shelf and deep zone. Underwater vegetation likely accounts for this unusual spectral signature and thus it is unsurprising that this lake required a unique band ratio to accurately tune the model. Measurements at

the second lake accounted for the shallowest range of depths of any lake (0.2 m - 2.1 m), which may have led to stronger reflectance in the red band, as the sand was more prominent.

In addition, an exponential relationship was able to better model depth ranges that include shallow depths of around 0.8 m, an outcome that is likely the result of incomplete transect sampling rather than physical significance. Of the three lakes at which a linear function provided the best model, two were the lakes at which depths on the littoral shelf were not measured; the third lake contained only a single measurement of the littoral shelf. Therefore, the lakes best modeled with a linear growth relationship are associated with measured bathymetry profiles that do not contain sufficiently shallow littoral shelf depths. This is evidenced by the prediction of negative depths at littoral shelves when applying linear models, the product of the strongly negative y-intercepts that render low spectral signature ratios negative. This leads us to conclude that the linear relationship between band ratios and depths at these lakes is more likely the product of the locations at which data was gathered rather than a result of physical significance. It is thus important to tune models to all regions (and all depths) of the lake.

3 Results

We produced bathymetry maps for 17 lakes on the ACP (Fig. 6), however the accuracy of these maps varies by lake and by depth. The best model variants for individual lakes at which depth data were collected were able to account for 58.5% - 97.6% of depth variability (median $R^2 = 0.86$, mean $R^2 = 0.82$; Table 3). Regional-scale models, however, were able to accurately explain less than half of the regional depth variability. Median uncertainty of single lake depth models (based on RMSE) was 1.23 m, while the average RMSE of the models was 1.44 m. However error was not distributed equally across depths and bathymetry rasters tend to represent a more limited range in depth than the measured depth points (Table 1, Table 3). In general, models tended to overestimate shallower depths and underestimate deep depths (Fig. 4d-f). When considering model-predicted depths at all study lakes, depth points less than 2.95 m were overestimated by an average of 0.21 m (or 17.2% of their true depth), with 61.3% of depths in this shallow-water group experiencing some model over-prediction. Meanwhile, 66.9% of depths greater than 2.95 m were underestimated, with an average difference between measured and modeled depths of 0.97 m. On average, points deeper than 2.95 m were underestimated by 5%. The threshold of 2.95 m represents the intersection between the 1:1 line and the correlation between measured and predicted depths. To address the underestimation of deep depths and overestimation of shallow depths in our models, additional transformations must be made, a goal that is outside the scope of this work.

Bathymetry accuracy variability by depth is at least partially explained by the fact that lake depth points are skewed heavily towards shallow depths, with approximately half of the data points representing depths less than 2.95 m. Only about 15% of the data points represented depths above 10 m. This is a function of the generally shallow nature of lakes on the Arctic Coastal Plain and the large area covered by littoral shelves within most study lakes (as seen on satellite imagery). Because of the relatively small number of deep water depth points, models were able to map bathymetry less accurately at deep central basins and therefore the bathymetry maps contain underpredicted deep water depths. In contrast to the skew in depth points, lakes were evenly divided into shallow

and deep classes. 9/17 lakes had some measured depths > 10 m and all of the study lakes had measured depths < 2.2 m (Table 1).

       Lake volumes ranged from 1.056 $\times 10^{-3}$ km$^3$ at the smallest lake (total surface area = 1.089 km$^2$) to 57.416 $\times 10^{-3}$ km$^3$ at the largest lake (total surface area = 18.998 km$^2$), with a median volume of 7.20 $\times 10^{-3}$ km$^3$ (Table 4). Volume and surface area were strongly correlated ($R^2 = 0.90$) for the 14 lakes at which complete volumes could be

modeled (Fig. 7). Linear models predicted negative depths across much of the lakes' shallow littoral shelves; thus, the modeled volumes of the three lakes at which linear models produced the most accurate results are an incomplete representation of the lake's water storage. Pixels at which models predicted negative depths were reclassified to a secondary NoData value of -1 and ignored when calculating water volume (i.e., water volume was calculated for the surface area with predicted depths greater than zero [Fig. 8]). Ground truth lake volume data do not exist for the

study lakes at a similar scale of analysis, rendering error metrics unfeasible (aside from those implicitly contained in the depth model error).

**4 Discussion**

4.1  Depth analysis

Our measured depth points capture the deep water depths on the ACP that many other studies neglect. Furthermore, depth was accurately derived from Landsat OLI imagery for individual lakes (the average $R^2$ value of the selected models for each lake was 0.82). Our $R^2$ values are consistent with those found in the literature (e.g. Jagalingam et al., 2015; Stumpf et al., 2003) and thus our selected models and derived maps can be considered successful. The regional scale model, however, was unsuccessful and regional volume analysis and mapping was

rejected.  This lack of model portability between lakes may be due to the fact that the blue band, most useful in determining depth, is the most susceptible to contamination from atmospheric aerosols. This finding is consistent with Smith and Pavelsky (2009), who found surprisingly high variability in a collection of remotely sensed lake storage volumes on the Peace-Athabasca Delta, Canada, despite their having similar physiographic setting and morphology.


4.2  Limitations

       The depth estimates are only tuned to the extrema of depths measured at each lake. Although gathering data across a lake's full bathymetric profile was attempted, it is likely that the depth minima and maxima were not captured at all lakes. Collecting data with sonar attached to a float plane limited the measurement of depths

approaching 0 m. Few pixels were sampled at the minimum depth that was able to be measured (0.2 m) and thus there is insufficient tuning to accurately model the littoral shelves of lakes. Furthermore, while we attempted to gather depths across the deep central basins, it is impossible at present to know whether we sampled the deepest point without measuring the entirety of the basin. Thus, depth maps may not accurately depict a lake's maximum depth.

The limited spatial resolution of Landsat imagery, in comparison with sonar depth data, constitutes the primary limitation to this work. As depths had to be averaged to conform to the assumption that each spectral

signature corresponded to a discrete depth, the spatial resolution and depth precision of the sonar depths was greatly degraded, potentially accounting for some of the inaccuracies in the model variants. Modeling bathymetry with satellite imagery of a higher spatial resolution would allow for the use of more training points and thus likely improve the accuracy of depth and volume predictions. Furthermore, samples were taken at a small fraction (in terms of surface area) of the lake (i.e. the entire lake's bathymetry was not mapped, rather, data points were collected along discrete and irregular transects). Thus, there exists a mismatch regarding the validation data and the natural phenomenon being modeled. Data at such a small spatial scale can never confirm with total accuracy the detailed nature of lake bed bathymetry. Constrained by cost and time however, collecting data at 17 remote lakes is an important step towards understanding Sand Sea lake bathymetries on Alaska's Arctic Coastal Plain.

### 4.3 Implications and future directions

Lakes on the Pleistocene Sand Sea may be categorized based on depth, littoral substrate, and water clarity, as seen in the study lakes, with such categories providing candidates for different model variations. Future projects may use this work to semi-automatically derive depths across the region, first manually classifying target lakes, and then applying different model variations to each class. Furthermore, subregions of each lake (e.g. deep basins, shallow shelves) may be classified in future studies and a different model variant applied to each subregion (e.g. variants that incorporate the red band applied to littoral shelves). Methods of lake subregion differentiation may include either (1) manual delineation based on spectral signatures or (2) automatic delineation with the aid of synthetic aperture radar (SAR) to determine regions of floating versus bedfast ice (which correspond with deep and shallow water, respectively; demonstrated by Engram et al., 2018; Jeffries et al., 1996). Additional future work may include validation of lake water volumes as additional bathymetric datasets become available.

### 5 Conclusion

This work provides a unique in situ depth dataset for lakes on the ACP and leverages these data alongside satellite remote sensing to map lake bathymetries and estimate volume. Lake volumes can be monitored using remote sensing, however at least one field visit must be made in order to select the best model for a given lake. As of yet, it is still challenging to universally model the bathymetry of lakes across northern Alaska. Instead, field data continues to be necessary to train and calibrate models on a per-lake basis.

Furthermore, lake morphology may evolve in glaciated regions such as northern Alaska in response to hydroclimatic changes and permafrost degradation (Arp et al., 2011, Liljedahl et al., 2011, Nitze et al., 2017). This implies that individual field surveys and static modeling efforts such as this one may not accurately represent ground conditions *ad infinitum,* particularly in the presence of a rapidly warming Arctic climate (Nitze et al., 2017). In addition to the persistent need for field data to address modeling limitations to spatial scale, field data collection and/or dynamic models will be important components if we are to model bathymetry across a longer temporal scale.

Despite these limitations, the simplicity of the depth modeling and bathymetry mapping approach has important benefits. The models can be tuned very rapidly and require relatively few data points for training in comparison to machine learning models (e.g., Sagawa et al., 2019), a useful feature when training data must be

collected in a relatively inaccessible region such as northern Alaska. In addition, the comparative nature of the
demonstrated modeling facilitates analysis of individual lake characteristics. Overall, this work provides an effective
dataset and methodology for mapping bathymetry of individual lakes in a unique geologic setting on the ACP.

**6 Data Availability**

We present a dataset to greatly increase the number of in situ measurements of lake depth on the little-
studied Inner Arctic Coastal Plain of Alaska. The dataset contains 13,735 point measurements of bathymetric depth
measured across 19 lakes, and is freely available through the National Science Foundation Arctic Data Center:
https://doi.org/10.18739/A2SN01440 (Simpson and Arp, 2018). The second dataset created for this project is
comprised of 17 bathymetry rasters, one for each lake at which a sufficient number of depth points was collected.
These rasters represent the depth predictions of the best performing model for each individual lake and are also
freely available through the National Science Foundation Arctic Data Center:
https://doi.org/10.18739/A2HT2GC6G (Simpson, 2019).

**Author contributions**

Claire E. Simpson and Christopher D. Arp designed the sampling. Christopher D. Arp secured the funding
and instrumentation for field work. Claire E. Simpson, Christopher D. Arp, and Benjamin M. Jones conducted the
field work. Claire E. Simpson processed and analyzed the data and prepared the figures and tables. Claire E.
Simpson prepared the manuscript with contributions from all co-authors.

**Competing interests**

The authors declare that they have no conflict of interest.

**Acknowledgements**

This research was supported by the National Sciences Foundation, NSF Project #1560372, REU:
Understanding the Arctic as a System, NSF Project #1417300, ALISS: Arctic Lake Ice Systems Science
(www.arcticlakeice.org), NSF Project #1806213, and the Bureau of Land Management Arctic District Office.
Additional funding for this project was provided by the NASA Arctic-Boreal Vulnerability Experiment (ABoVE)
grant NNX17AC60A to L.C.S., and the University of California, Los Angeles Honors Program's Irving and Jean
Stone Research Award. Special thanks to Jim Webster for flight support.

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

**Table 1**: **Sampling specifications for each study lake**. The number of sample points and measured depth range were calculated after the points were processed for quality assurance (e.g. anomalous depth pixels removed) but before resampling to the single point per pixel dataset.

| Lake ID | Centroid latitude (dd) | Centroid longitude (dd) | Number of points sampled | Measured depth range (m) |
|---------|------------------------|-------------------------|--------------------------|--------------------------|
| 2964 | 70.3616 | -153.6750 | 703 | 1.5 - 11.8 |
| 3442 | 70.3168 | -153.8160 | 1065 | 0.8 - 2.4 |
| 3839 | 70.2875 | -153.8620 | 1401 | 1.6 - 6.1 |
| 4199 | 70.2457 | -154.4680 | 645 | 1.2 - 12.8 |
| 4222 | 70.2484 | -153.1460 | 1216 | 0.2 - 11.7 |
| 4291 | 70.2386 | -153.2030 | 479 | 0.2 - 6.2 |
| 4365 | 70.2304 | -153.2500 | 287 | 0.2 - 4.9 |
| 4782 | 70.1983 | -153.3150 | 870 | 0.2 - 2.6 |
| 5211 | 70.1581 | -153.9300 | 762 | 0.7 - 5.0 |
| 5242 | 70.1585 | -154.2260 | 745 | 1.9 - 8.3 |
| 5326 | 70.1349 | -154.1280 | 656 | 0.6 - 4.4 |
| 5570 | 70.0948 | -153.7480 | 392 | 0.8 - 10.4 |
| 5893 | 70.0577 | -153.5010 | 1113 | 0.3 - 21.3 |
| 6058 | 70.0285 | -153.3670 | 284 | 0.6 - 17.0 |
| 6167 | 70.0122 | -153.0930 | 1715 | 0.8 - 14.6 |
| 6199 | 70.0110 | -153.4720 | 991 | 2.1 - 10.7 |
| 6274 | 69.9967 | -153.0690 | 280 | 0.3 - 13.2 |


**Table 2: Equations for Modeling Depth.** Modeled depth (Z) is calculated with each of four equations that are tuned with each of three input band pairs. $R_i$ and $R_j$ represent the Top-of-Atmosphere reflectances of bands i and j, respectively. Band pairs (band i/band j) include the blue and red bands, the blue and green bands, and the green and red bands. Tunable parameters $m_1$ and $m_0$ are derived by comparing spectral signatures with depth (as in Fig. 4a-c).

| | | Ratio adjustment type | |
|---|---|---|---|
| | | Transform | Simple |
| Growth factor | Linear | $Z = m_1\left(\dfrac{\ln(nR_i)}{\ln(nR_j)}\right) - m_0$ | $Z = m_1\left(\dfrac{R_i}{R_j}\right) - m_0$ |
| | Exponential | $Z = m_0 e^{m_1\left(\frac{\ln(nR_i)}{\ln(nR_j)}\right)}$ | $Z = m_0 e^{m_1\left(\frac{R_i}{R_j}\right)}$ |


**Table 3**: **The best spectral depth model for each lake (based on $R^2$).** A simple ratio exponential function provided the best model for the greatest number of lakes, while the blue/green and blue/red band ratios both provided good inputs for models at different lakes, accounting for the best spectral-depths models at 8 and 7 lakes, respectively. The average $R^2$ of the best model at each lake is 0.818, with an average root mean squared error (RMSE) of 1.439 m. [1]Some suspended sediment is visible, however it does not overlap the area at which depths were measured.

| Lake ID | Modeled depth range (m) | Turbid? | Best method | | | $R^2$ | RMSE |
|---|---|---|---|---|---|---|---|
| | | | Ratio type | Band ratio | Growth type | | |
| 2964 | 1.6 – 11.7 | No | Transform | Blue/Green | Linear | 0.802 | 1.973 |
| 3442 | 0.9 – 1.8 | Yes | Simple | Green/Red | Exponential | 0.916 | 0.270 |
| 3839 | 1.4 – 5.1 | Yes | Simple | Blue/Red | Exponential | 0.871 | 0.600 |
| 4199 | 0.6 – 8.2 | No | Simple | Blue/Green | Exponential | 0.632 | 2.132 |
| 4222 | 0.4 – 8.9 | Yes | Simple | Blue/Red | Exponential | 0.689 | 2.307 |
| 4291 | 0.8 – 4.5 | Yes | Simple | Blue/Red | Exponential | 0.585 | 0.835 |
| 4365 | 0.2 – 3.7 | No | Simple | Blue/Green | Linear | 0.784 | 0.478 |
| 4782 | 0.4 – 2.2 | Yes | Transform | Green/Red | Exponential | 0.893 | 0.138 |
| 5211 | 0.8 – 4.7 | Yes | Simple | Blue/Red | Exponential | 0.804 | 0.563 |
| 5242 | 2.1 – 7.2 | Yes[1] | Simple | Blue/Green | Exponential | 0.976 | 0.684 |
| 5326 | 0.8 – 4.4 | No | Simple | Blue/Red | Exponential | 0.862 | 0.425 |
| 5570 | 1.1 – 6.4 | No | Simple | Blue/Green | Exponential | 0.654 | 1.957 |
| 5893 | 0.5 – 21.1 | Yes[1] | Transform | Blue/Green | Exponential | 0.954 | 1.931 |
| 6058 | 0.6 – 9.9 | No | Simple | Blue/Green | Exponential | 0.866 | 3.568 |
| 6167 | 0.3 – 11.1 | Yes[1] | Transform | Blue/Green | Exponential | 0.848 | 2.604 |
| 6199 | 1.0 – 9.5 | Yes | Simple | Blue/Red | Linear | 0.907 | 1.227 |
| 6274 | 0.3 – 8.8 | Yes[1] | Simple | Blue/Red | Exponential | 0.867 | 2.765 |

**Table 4: Modeled Lake Volumes.** Individual lake volumes were estimated by multiplying the modeled depth for each pixel by a constant factor of 900 m$^2$ (Landsat spatial resolution). Depths were modeled by applying the best spectral-depth model for the lake (Table 3). Linear depth models predicted negative depths for some pixels; volume estimates derived from such models (namely the models applied at lakes 2964, 4365, and 6199) include only those pixels with modeled depths greater than zero. The percent of the surface area for which depth estimates at a lake were positive (in contrast to the total surface area of a given lake derived using the NDWI mask) is quantified.

| Lake ID | Total surface area (km$^2$) | Surface area with depths modeled (% total area) | Modeled volume ($10^{-3}$ km$^3$) |
|---|---|---|---|
| 2964 | 4.631 | 25.78 | 11.113 |
| 3442 | 1.089 | 100 | 1.056 |
| 3839 | 5.419 | 100 | 10.367 |
| 4199 | 1.953 | 100 | 3.454 |
| 4222 | 1.533 | 100 | 8.371 |
| 4291 | 0.637 | 100 | 2.229 |
| 4365 | 1.046 | 61.66 | 2.102 |
| 4782 | 6.455 | 100 | 7.202 |
| 5211 | 9.865 | 100 | 19.280 |
| 5242 | 18.998 | 100 | 57.416 |
| 5326 | 4.846 | 100 | 10.545 |
| 5570 | 0.913 | 100 | 2.464 |
| 5893 | 10.552 | 100 | 37.949 |
| 6058 | 1.559 | 100 | 6.336 |
| 6167 | 2.778 | 100 | 10.343 |
| 6199 | 2.038 | 37.24 | 4.943 |
| 6274 | 0.662 | 100 | 2.484 |





**Figures**

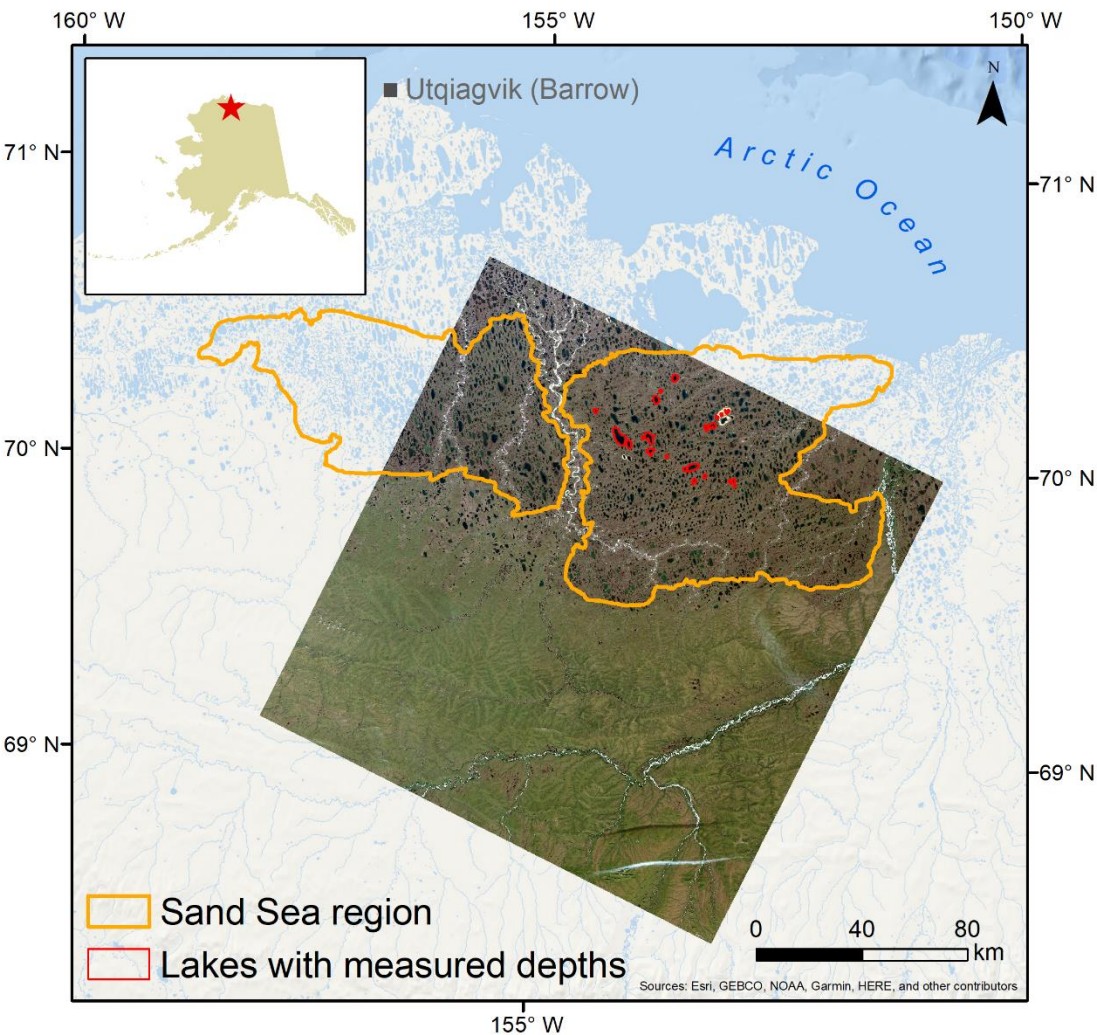


**Figure 1.** The lake-rich area of interest on Alaska's Arctic Coastal Plain (ACP), southeast of Utqiaġvik (Barrow). The imagery used in our models is a Landsat-8 tile (Path 077, Row 011) acquired on 5 August 2016. The Pleistocene Sand Sea, a geologically-unique region of the ACP, is delineated based on a classification of eolian sand by Jorgenson et al., 2014. Landsat-8 image courtesy of the U.S. Geological Survey.

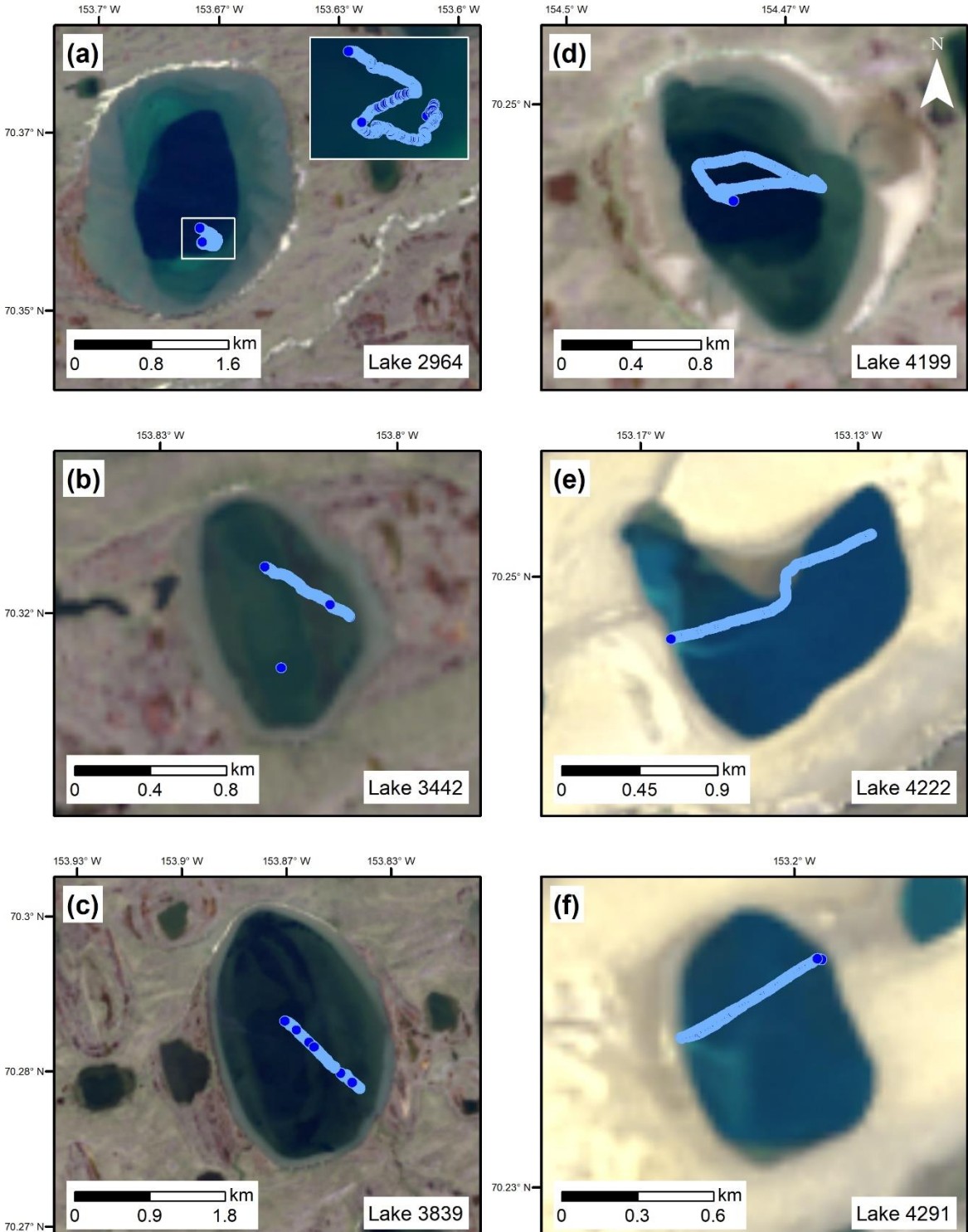


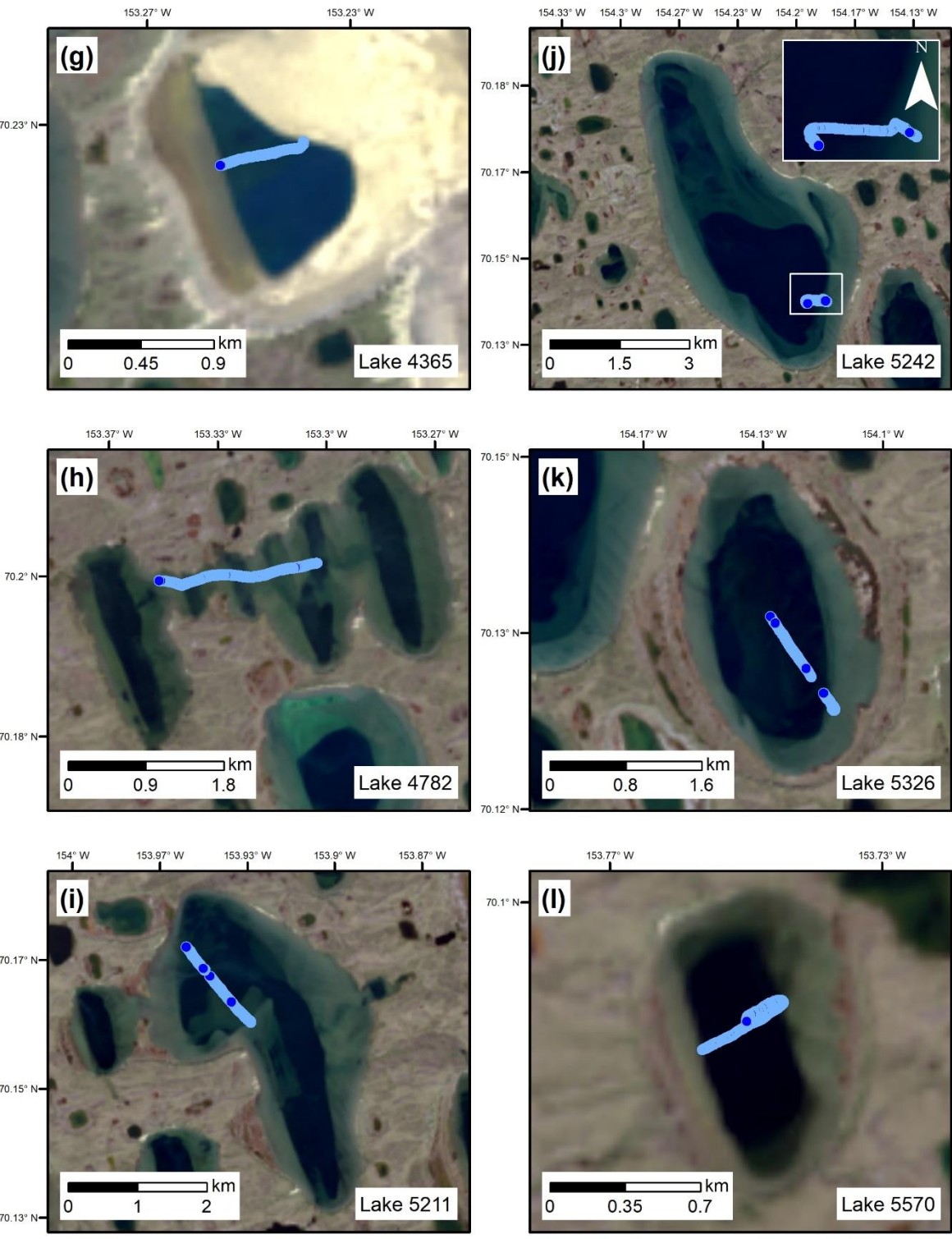

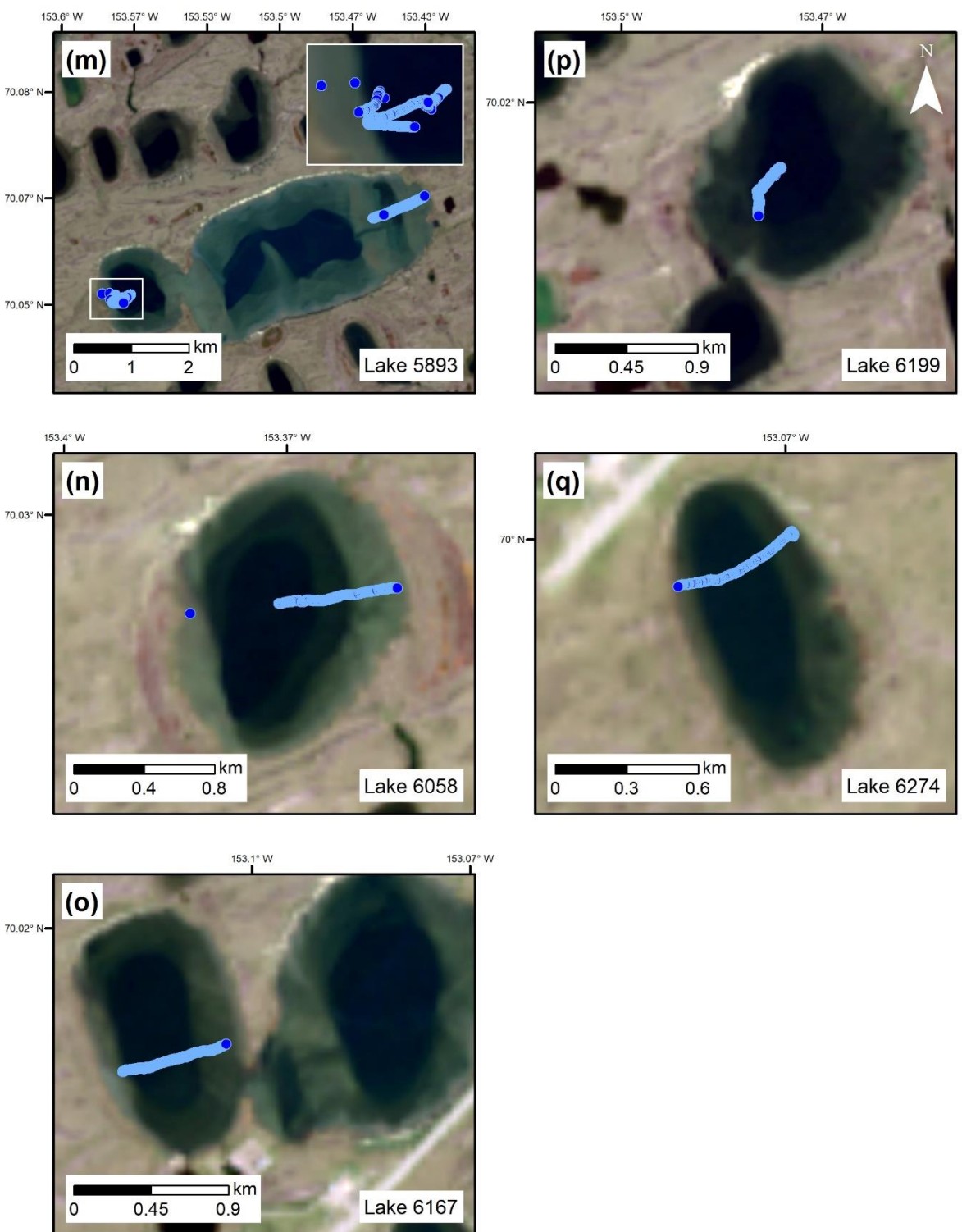

**Figure 2.** Transects were measured across 17 study lakes. Although the transects follow irregular paths (due in part to wind conditions and sonar error), all but two of the transects capture a range of depths from the deep central basins to the shallow outer shelves. These are the full transects before resampling to a single point per pixel. Where the form of a transect is unclear, inset maps are provided. Landsat-8 image courtesy of the U.S. Geological Survey.

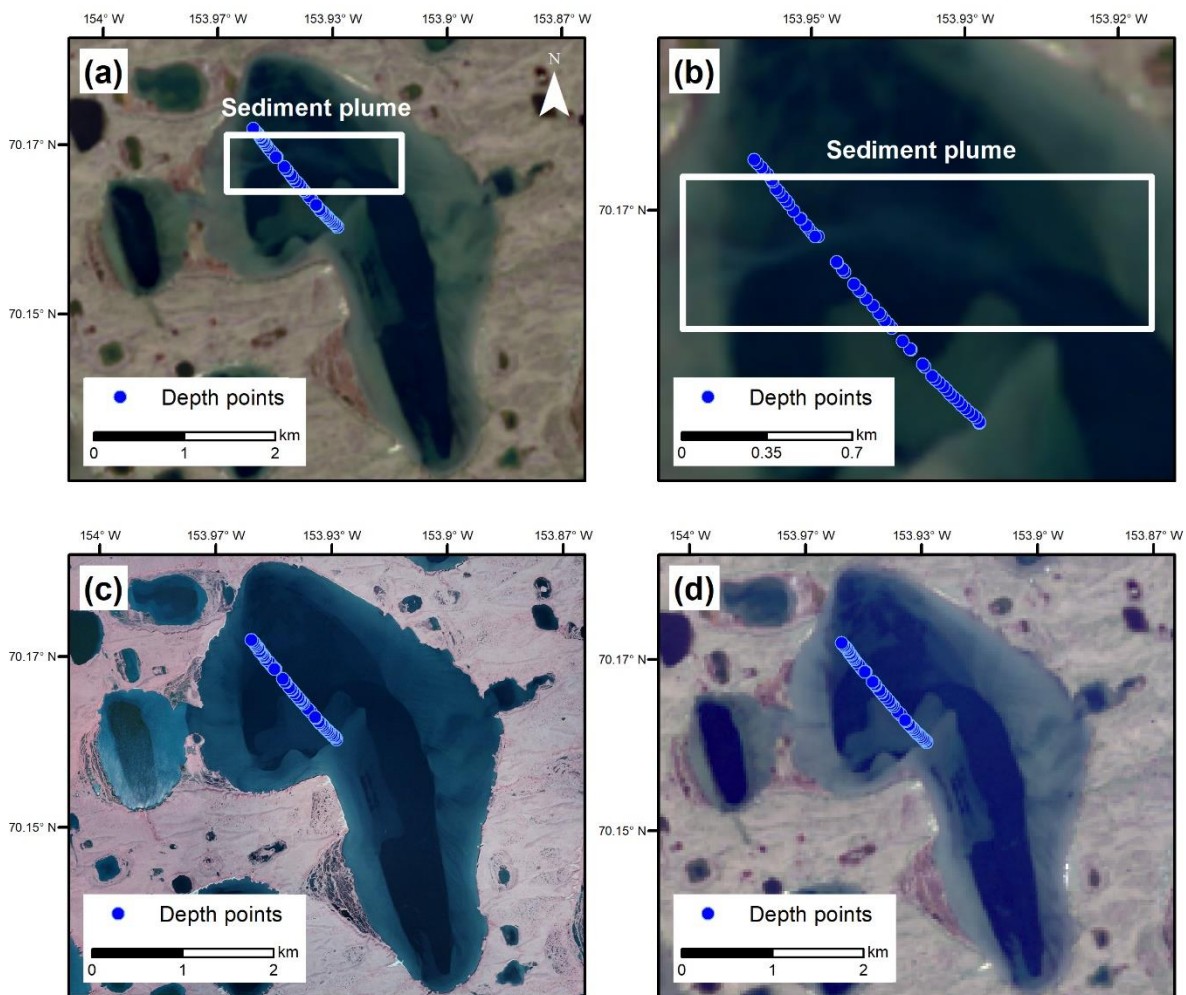

**Figure 3.** Sediment is detected in RGB Landsat imagery (acquired 5 August 2016) of a representative study lake **(a,b)**. This is confirmed as a temporary sediment plume by comparing the image of the lake used in modeling to 2.5-m color-infrared photography acquired 18 July 2002 **(c)** and a Landsat image acquired 13 July 2016 **(d)** in which no sediment plumes are visible. Landsat-8 images and Digital Orthophoto Quadrangles (DOQs) courtesy of the U.S. Geological Survey.

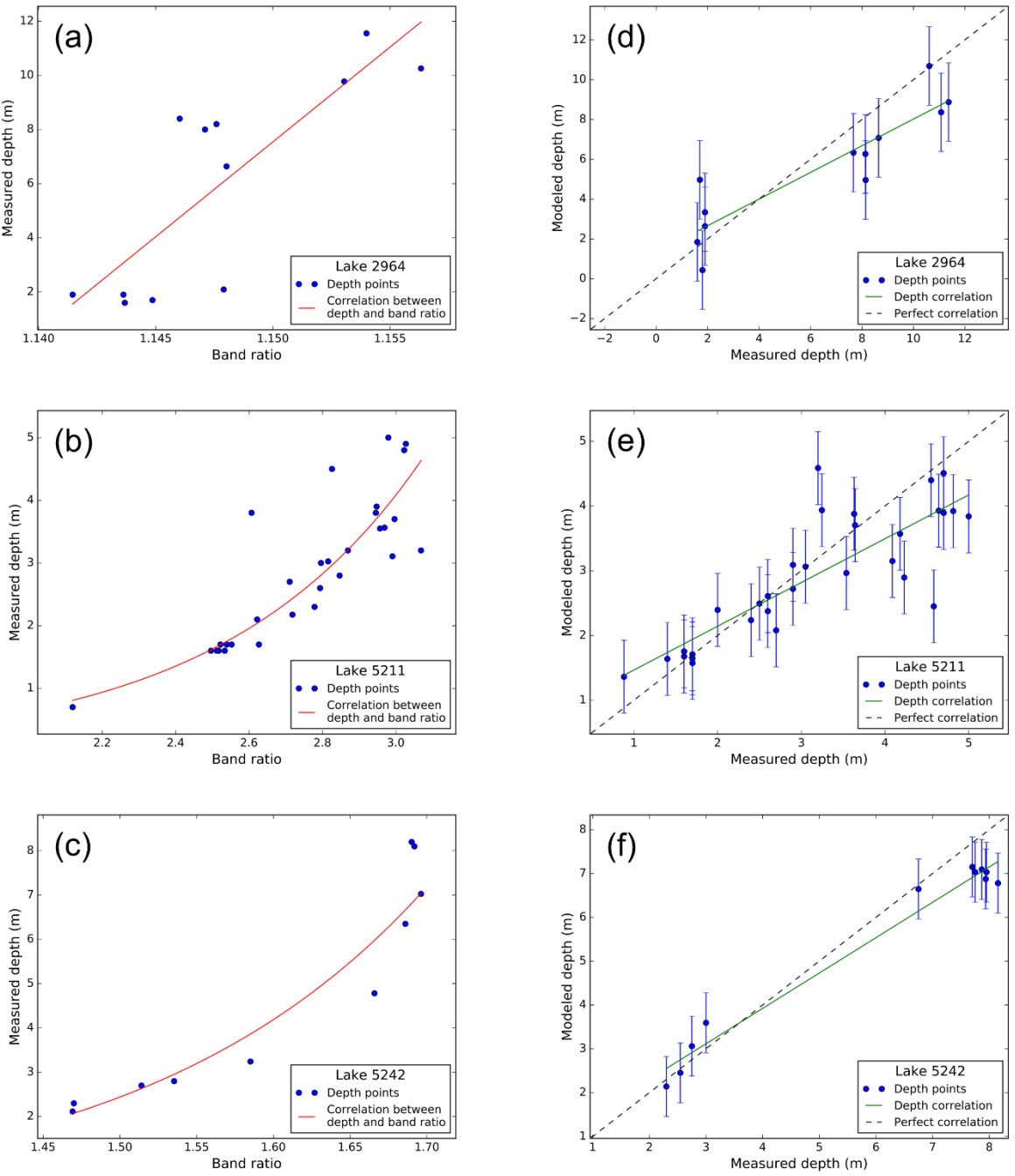

**Figure 4.** Coefficients of the trendlines between band ratios and measured depths **(a-c)** are used to tune the depth models for each lake. Different models (specified for each lake in Table 3) best predicted lake depth at each of these three lakes. Correlation between measured and modeled lake depths at three representative lakes **(d-f)** reveals underestimation of deeper depths and overestimation of shallow depths. Error bars represent root mean squared error (RMSE).

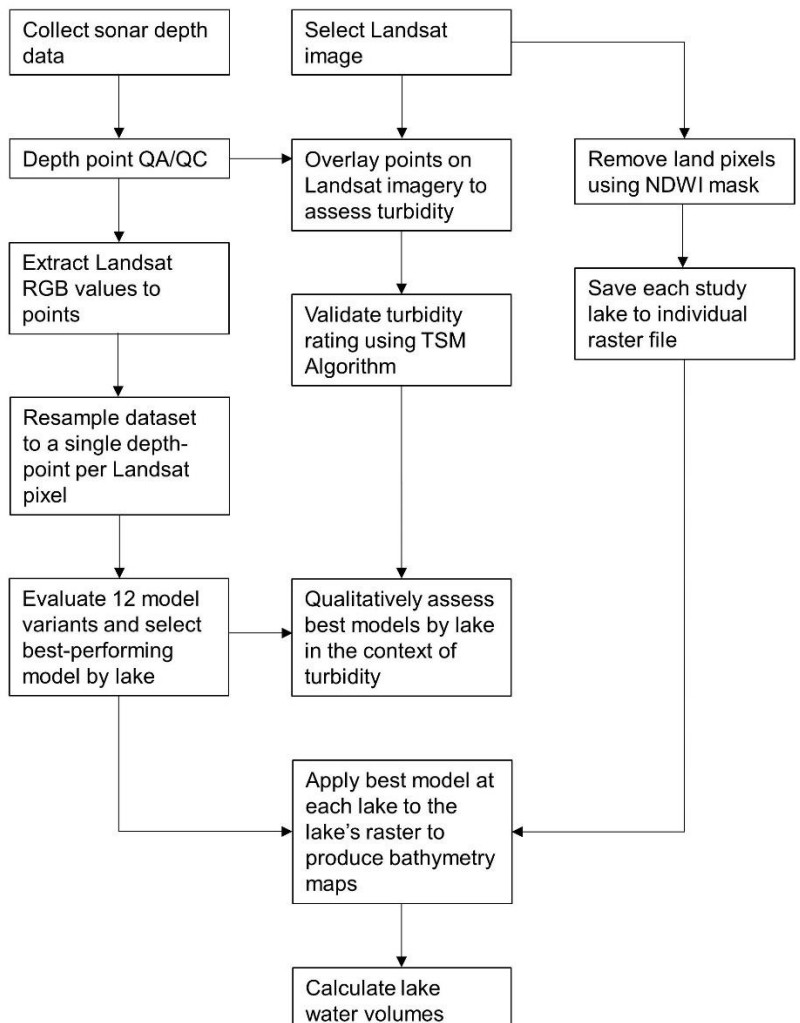

**Figure 5.** Sequence of processing, analysis, and production steps used to map bathymetry and derive lake water volumes with depth points and Landsat imagery.

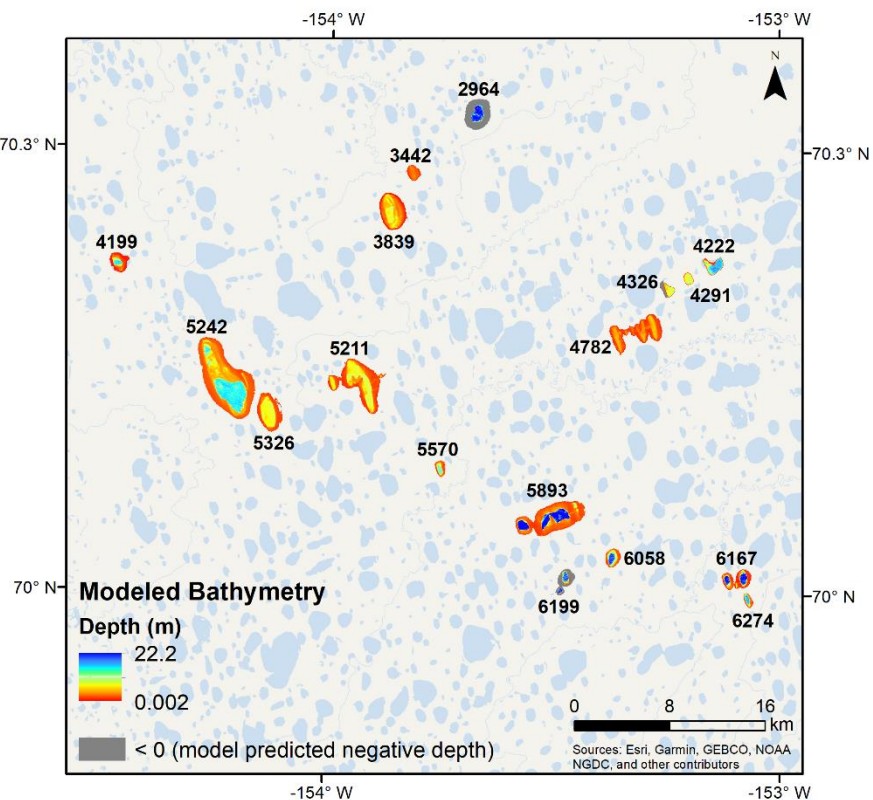

**Figure 6.** Bathymetry was modeled individually for each study lake and all bathymetry rasters were ultimately mosaicked together. The color bar indicates the depths predicted by the model variants at each lake; grey represents the pixels at which negative depths were modeled (these negative depths have been reclassified to -1 in the published bathymetry raster dataset, Simpson, 2019).


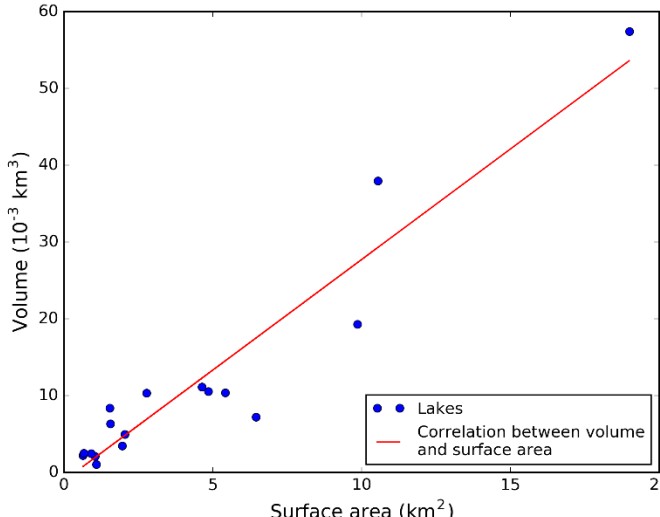

**Figure 7.** A strong correlation exists between surface area and modeled volume for the 17 lakes we analyzed.


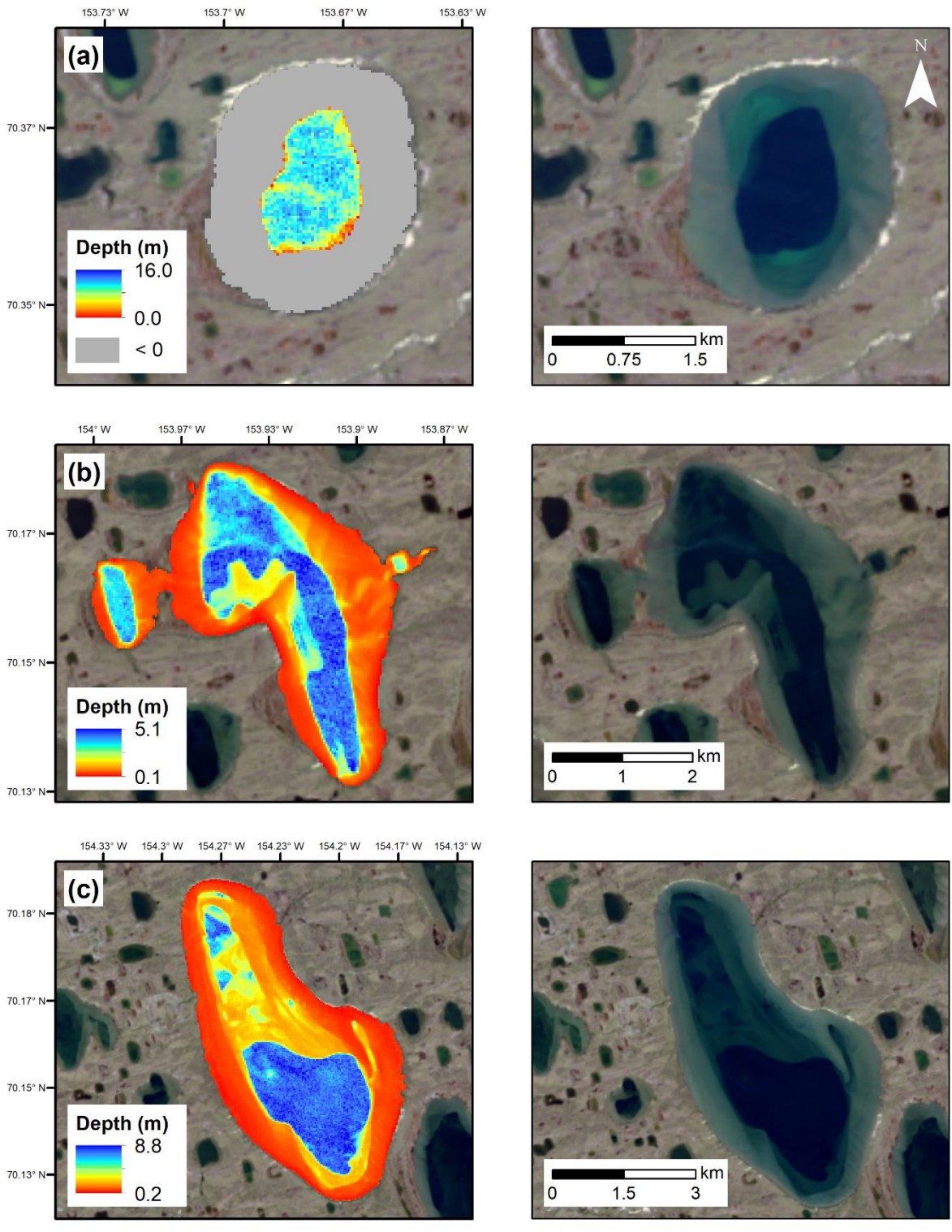

**Figure 8.** Modeled lake bathymetry at a representative lake (**a**) reveals the tendency of linear depth models to drastically underestimate the depths of the littoral shelves when not calibrated to shallow depths. Conversely, the exponential depth models

applied to other lakes are promising across both littoral shelves and central basins (**b**, **c**). The products of three different spectral-depth model variations are overlain on the Landsat imagery from which the products were derived. Adjacent to each depth product is the original Landsat imagery of the lake. Color bars indicate the depths predicted by the model variants at each lake, while the grey area **(a)** represents the pixels at which negative depths were modeled (these negative depths have been reclassified to -1 in the published bathymetry raster dataset, Simpson, 2019). Landsat-8 image courtesy of the U.S. Geological Survey.





