# Peer review of "Landsat-derived bathymetry of lakes on the Arctic Coastal Plain of Northern Alaska"

_Earth System Science Data, 2019_

## Short Comment (SC1) · 18 Feb 2020

Several steps of the depth data processing and Landsat image selection, processing, and analysis sections, e.g. eliminating lakes with insufficient measurements, removing anomalous values, and visually inspecting lakes suggests several caveats for how depth point data was selected for analysis. Some additional explanation for the impact data would have had on the model had it not been left out could be beneficial. While acknowledging the constraints of using the Landsat-8 spatial resolution in conjunction with the sonar depth points means the depth point data must be degraded, further explanation of the methodology for processing and analyzing the depth point data could be helpful to bolster the methods section.

---

## Referee Comment (RC1) · Ali P. Yunus (Referee) · 13 Apr 2020

The manuscript "Remote sensing of lake water volumes on the Arctic Coastal Plain of Northern Alaska" present an important lake bathymetric dataset across the Arctic Coastal Plain using sonar depth data and modeled lake bathymetry raster's from the Landsat images. It will be served as a baseline for the further studies, such as changes in water resources, energy balance, and ecological habitat. Although the novelty of methodology is comparatively low, but the dataset is of good use and paper was well organized and written. But there are still a lot of issues that need to be resolved before publishing in ESSD. General Comments The authors have measured 17 lakes using the sonar instrument in the study region. However very little is detailed on the measurement part. For example, there should be a table highlighting the number of points mea-

sured in each lakes, what is the minimum measured depth, maximum depth measured, etc. Indeed, this can be incorporated in Table 3. One another major concern is about the dataset itself. There is a little information provided on how the authors carried the field survey such as criteria on selection of transects in each lake (how many transects, which direction). I can see that authors pointed that a depth range from the littoral shelf to the deep central basin was captured. However, since this is a data paper, I would like to see all the lake transects in a Figure format (similar to Figure 2a). This won't be difficult as there are only 17 lakes are studied. Along with this figure, one photograph showing the sonar instrument mounted on the platform will be useful for readers to visualize the process. Authors used Landsat images of 2016 in this study for modeling the bathymetry. Why didn't the authors considered Sentinel-2 images of 2017 July -August, which is having higher spatial resolution than Landsat in this study?. This should be clarified. As mentioned in line 110, authors used TOA reflectance's, which is different from water leaving reflectance. That means, no atmospheric correction algorithm is performed in this work. USGS/Earthexplorer directly providing the atmospherically corrected reflectance products, apart from various other atmospherically corrected algorithms available such as simple DOS, FLAASH, ACOLITE, iCOR etc. Don't the authors believe that atmospherically corrected images may improve the accuracy of predictions as it corrects the haze and specular reflections?. This should be clarified. Refer Vanhellemont, Q., & Ruddick, K. (2015). Advantages of high quality SWIR bands for ocean colour processing: Examples from Landsat-8. Remote Sensing of Environment, 161, 89–106. https://doi.org/10.1080/22797254.2018.1457937 Line 162, "Study lakes were then visually assessed to provide a Boolean turbidity rating for...". How to validate this ?. A suggestion is may be used some well known TSM algorithm such as that in Acolite built one (TSM_Nechad) and compare the concentration of TSM in 17 lakes. Validation of the modelled bathymetric results can also be performed by drawing profiles comparing modelled vs measured transects (transects used in training cannot be used for validation). See AP. Yunus, Jie Dou, Xuan Song, Ram Avtar. (2019). Improved Bathymetric Mapping of Coastal and Lake Environments Using Sentinel-2 and

Landsat-8 Images. Sensors 2019, 19(12), 2788; https://doi.org/10.3390/s19122788 Specific Comments Section 2 should be Data and Methods Line 202. "The best models for lakes at which. . .". Mention which one is the best model here. Line 239. "Depth was accurately derived from Landsat OLI imagery for individual lakes". How much accurate. Accuracies should be clearly provided in quantified values using RMSE, ME etc. Lake bathymetry is continuously reworked, particularly in glaciated regions. This should be discussed in the light of additional field surveys.

---

## Referee Comment (RC2) · Ingmar Nitze (Referee) · 2 Aug 2020

The authors present the methods and data to derive lake depths from Landsat imagery for the Sand Sea region of the Arctic Coastal Plain in Northern Alaska. The lakes in this regions are characterized by a special morphology with distinct shelves and typically deep central basins. Compared to other lakes on the ACP these lakes are very clear and therefore potentially suitable for water depth estimation with optical remote sensing data. The topic and scientific problem is important for understanding lake related landscape processes and associated activities in permafrost. For this region this study is unique and important for the understanding of permafrost hydrology. However, probably the scope is rather local and specific for ESSD. The presentation of the manuscript is good, but needs some more detail in several parts. These include e.g.

[Figure]

the raw data acquisition and software/workflow, preferentially with processing code. It is unclear which software the authors used for data processing/model creation. The analysis is robust, but rather basic, with only one used image, simple band ratios and simple regression models. The authors may apply more advanced analysis in all these points, which are easily accessible in several software packages of choice (e.g. python with scikit-learn). They would probably help create a regional model. The inclusion of software/platform used, would have helped to provide more specifics. The datasets are accessible and working.

Detailed comments and suggestions for improvement are stated below.

54: supra lake bluffs?

95ff: rather undetailed description.

95ff: Transect lines how dense? Which observation frequency distance between points? I like the method to use a float plane as a platform for taking samples. I think this is very important for fieldwork in the Arctic.

106: 2.2. Your database of only one used image is VERY small. I suggest that you use more different images. There should be plenty of data and acquisitions

107: Does it make sense to merge them initially? I think it is ok to do that later or on the fly, but differences between lakes would be an interesting point to analyze. Spatial sampling strategies would help to make the analysis more robust.

109: TOA data OK as SR is not perfect in high latitudes. Also OK for only one image. Please refer that there are SR data available, but that they come with disclaimer for high latitudes.

111: Interesting that the coastal band did not improve results as it is specifically designed for this purpose as far as I know.

124: maybe you can use a better term than degraded. E.g. merged, resampled

142: The statement that there was no suitable 2017 image should come earlier, as it is confusing to start stating that you used 2016 imagery.

162: This sentence sounds a bit unclear and complicated to me.

187: You mentioned that you merged all data, here you say you have them sampled for each lake. Please clarify.

187: I am not sure is a random sampling within one lake is really feasible, as spatial autocorrelation can be an issue. This may lead to an overestimation of your accuracy. However, I understand that input data were taken as a transect and that field work in the Arctic often prohibits more sophisticated/robust spatial sampling design (e.g. grid).

245: If blue performs well, then the question is why the "coastal" band did NOT perform well, as it is more or less designed for this application.

281: good point mentioning the limitation of extrema. I would say this is typical for almost all predictive models.

301 ff: I would like to see a discussion about regression models. Are there better models available? More sophisticated one, which take more information into account? E.g. Machine-learning or Deep-Learning models, LASSO, . . .

Fig 5a: Should "< 0" be "> 0"?

---

## Author Comment (AC1) · 9 Sep 2020

We appreciate the reviewers' constructive comments, which we have incorporated in this revision. As a result, the quality of this manuscript has been substantially improved. We have listed reviewer comments and addressed them line-by-line below.

RC1:

1. The authors have measured 17 lakes using the sonar instrument in the study region. However very little is detailed on the measurement part. For example, there should be a table highlighting the number of points measured in each lakes, what is the minimum measured depth, maximum depth measured, etc. Indeed, this can be incorporated in Table 3.

[Figure]

a. This comment has been taken. We have incorporated the number of points measured per lake into a new Table 1, which also contains maximum and minimum depths measured. We have also added more information in Section 2.1, including information about the frequency of sampling.

2. There is a little information provided on how the authors carried the field survey such as criteria on selection of transects in each lake (how many transects, which direction). we can see that authors pointed that a depth range from the littoral shelf to the deep central basin was captured. However, since this is a data paper, we would like to see all the lake transects in a Figure format (similar to Figure 2a). This won't be difficult as there are only 17 lakes are studied

a. We have added additional information about the transects, including information about the important features of transects in Section 2.1: "It should be noted that, as transects were comprised of individual points whose relationship to one another was unimportant to the modeling, the direction, angle, and other qualities of the transect are significantly less important than the range of depths captured". We have also created the suggested figures showing transects.

3. One photograph showing the sonar instrument mounted on the platform will be useful for readers to visualize the process.

a. Unfortunately, we do not have a good image of this.

4. Authors used Landsat images of 2016 in this study for modeling the bathymetry. Why didn't the authors considered Sentinel-2 images of 2017 July -August, which is having higher spatial resolution than Landsat in this study? This should be clarified.

a. This was a pilot project and at the time that we started we had ready access to Landsat data but did not have easy access to Sentinel. Redoing the analysis with Sentinel could potentially improve the fidelity of the results but is beyond the scope of the project at this time.

5. As mentioned in line 110, authors used TOA reflectance's, which is different from water leaving reflectance. That means, no atmospheric correction algorithm is performed in this work. USGS/Earthexplorer directly providing the atmospherically corrected reflectance products, apart from various other atmospherically corrected algorithms available such as simple DOS, FLAASH, ACOLITE, iCOR etc. Don't the authors believe that atmospherically corrected images may improve the accuracy of predictions as it corrects the haze and specular reflections? This should be clarified.

a. We were partial to TOA reflectance rather than SR as SR algorithms are often suboptimal when looking at water bodies due to the low level of water leaving radiance. Additionally, SR corrections are not always accurate at high latitudes and can sometimes induce artifacts on water. We did compare the TOA reflectance and SR values at our sample locations and found correlations > 0.99 for the blue, green, and red bands. We have included information into Section 2.2 to explain why TOA was used.

6. "Study lakes were then visually assessed to provide a Boolean turbidity rating for. . .". How to validate this ?. A suggestion is may be used some well known TSM algorithm such as that in Acolite built one (TSM_Nechad) and compare the concentration of TSM in 17 lakes. a. We have incorporated this statistical analysis in the Results section to validate our visual method. As suggested, we used the ACOLITE implementation of the Nechad et al. TSM algorithm, which agreed with the majority of our qualitative turbidity assessments. We also included a discussion of the limits of this TSM analysis (namely that the algorithm was very sensitive to depth) and how we navigated around this fact to obtain relative TSM results for each lake.

7. Validation of modeled bathy results can be performed by drawing profiles comparing modeled vs measured transects (transects used in training cannot be used for validation).

a. Unfortunately with our data, graphs of this nature are visually noisy and hard to interpret. Our transects are not strictly 2-dimensional - they are not straight transect lines,

and instead can meander and double back across a lake. This presents a challenge when depicting transects on 2 dimensional graphs as compressing the information to simple distance across transect and depth values shows a distortion and consequentially the transects do not appear on the graph as smooth bathymetry profiles. We did try to visualize our transects on 3-dimensional plots, however these were cluttered and hard to interpret. We will also clarify that we trained and validated our data at the point level, rather than the transect level.

8. Section 2 should be Data and Methods

a. We have changed this.

9. Line 202. "The best models for lakes at which..". Mention which one is the best model here.

a. There is not one single best model, we are instead referring to the best model for each individual lake. We have edited the text to better convey this: "The best model variants for individual lakes at which depth data were collected..."

10. Line 239 "Depth was accurately derived from Landsat OLI imagery for individual lakes". How much accurate. Accuracies should be clearly provided in quantified values using RMSE, ME etc.

a. We have specified an accuracy metric in this line: "Depth was accurately derived from Landsat OLI imagery for individual lakes (the average R2 value of the selected models for each lake was 0.82)"

11. Lake bathymetry is continuously reworked, particularly in glaciated regions. This should be discussed in the light of additional field surveys.

a. In the conclusion, we have added a brief discussion of the potential for lake bathymetry to evolve in response to permafrost degradation or hydroclimate changes. This is mentioned in the context of the continuing need for field data and/or dynamic models to understand bathymetry across a longer timescale.

RC2:

General Comments:

1. The presentation of the manuscript is good, but needs some more detail in several parts. These include e.g. the raw data acquisition and software/workflow, preferentially with processing code. It is unclear which software the authors used for data processing/model creation.

a. We have added to the Data and Methods section, e.g. sampling frequency, figures showing transects across lakes, number of points collected at each lake, and noted our use of Python and ArcGIS.

2. The analysis is robust, but rather basic, with only one used image, simple band ratios and simple regression models. The authors may apply more advanced analysis in all these points, which are easily accessible in several software packages of choice (e.g. python with scikit-learn). They would probably help create a regional model. The inclusion of software/platform used, would have helped to provide more specifics.

a. We have added more information about software used. With regards to the simplicity of methods, there is another project in progress that addresses this question and tests some of these other recommended methods.

Line-By-Line Comments:

54: supra lake bluffs?

a. We have changed the wording to "bluffs surrounding lakes" to clarify.

95ff: rather undetailed description.

95ff: Transect lines how dense? Which observation frequency distance between points? I like the method to use a float plane as a platform for taking samples. I think this is very important for fieldwork in the Arctic.

a. We have added more information on this to Section 2.1

106: 2.2. Your database of only one used image is VERY small. I suggest that you use more different images. There should be plenty of data and acquisitions

a. There is very little quality imagery around our area of interest in the relevant time frame. Only a few months exist when our lakes of interest are ice-free and at a relatively similar lake level to the level at the time of in situ measurement (small seasonal fluctuations in lake level exist). Furthermore, many of the images that would have been potentially useful were cloudy and had negative atmospheric effects. Therefore it was difficult to find useful images so we decided to stick with the conventional image-specific modeling approach to obtain more accurate results from the one image rather than try to generalize. As this is a pilot project, we believe that using one image is sufficient for the scope of this work.

107: Does it make sense to merge them initially? I think it is ok to do that later or on the fly, but differences between lakes would be an interesting point to analyze. Spatial sampling strategies would help to make the analysis more robust.

a. The depth points were merged to promote efficient processing e.g. when extracting the spectral values, file reading/editing, and to help visualize them against our image of choice, however each point's lake ID was retained so no data was lost when the individual lake transect points were merged into a single file. Ultimately, we are still doing the bulk of the analysis at the single lake level.

109: TOA data OK as SR is not perfect in high latitudes. Also OK for only one image. Please refer that there are SR data available, but that they come with disclaimer for high latitudes.

a. We have added this information in Section 2.2

111: Interesting that the coastal band did not improve results as it is specifically designed for this purpose as far as I know.

a. agreed

124: maybe you can use a better term than degraded. E.g. merged, resampled

a. Changed to resampled

142: The statement that there was no suitable 2017 image should come earlier, as it is confusing to start stating that you used 2016 imagery.

a. We have added this statement to the sentence first stating the use of imagery from 2016.

162: This sentence sounds a bit unclear and complicated to me.

a. We have reworded to clarify.

187: You mentioned that you merged all data, here you say you have them sampled for each lake. Please clarify.

a. We have added this information to the first sentence in Section 2.2, hopefully this provides enough clarification.

187: I am not sure is a random sampling within one lake is really feasible, as spatial autocorrelation can be an issue. This may lead to an overestimation of your accuracy. However, I understand that input data were taken as a transect and that field work in the Arctic often prohibits more sophisticated/robust spatial sampling design (e.g. grid).

a. We have clarified that the sampling is semi-random at the lake level and fully random at the regional level. We were indeed limited in our collection of sample points by time, location access, equipment, etc.

245: If blue performs well, then the question is why the "coastal" band did NOT perform well, as it is more or less designed for this application.

a. This is a great question, this is a pilot project and therefore we stuck with existing methods of band incorporation, which, at the time of analysis, did not incorporate the

coastal band.

281: good point mentioning the limitation of extrema. I would say this is typical for almost all predictive models.

301 ff: I would like to see a discussion about regression models. Are there better models available? More sophisticated one, which take more information into account? E.g. Machine-learning or Deep-Learning models, LASSO, . . .

a. This is a great point, there are certainly more sophisticated models available, and we have been exploring some of these other methods for another project.

Fig 5a: Should "< 0" be "> 0"?

a. These are negatively predicted values due to use of a linear model tuned to less than the shallowest extremes. The representation of negative values is clarified in the figure caption.

---

## Author Comment (AC2) · 9 Sep 2020

We would like to extend our appreciation for this constructive comment.

We have added additional information about how the data was collected and processed and included a brief discussion regarding the impact on the models if some of the removed data had been used in training (e.g. at lakes with insufficient samples, we would see model overfitting; if the anomalous bright points at the margins of Pik Dunes lakes had been used in training, the model would have had to reconcile associating very different spectral signatures with comparable depths and therefore model performance would have likely decreased).

[Figure]

2020.

---

## Editor Decision (ED1)

Dear Authors and Colleagues

Thank you for your contributions
Thanks for the authors for the replies to the reviews of your paper and for the revision.

The published time series on lake bathymetry on the Arctic Coastal Plain of Northern Alaska is of importance. The manuscript, the data description and data publication do not yet fulfill the requirements of ESSD and improvements are needed. A major revision of the manuscript and a minor revision of the dataset is needed.

The manuscript provides a lot of detailed information on the generation of the Landsat-derived lake bathymetric data. However, the focus on the data sets is lost in the complexity of the paper. This happens because the article is not exclusively focused on the datasets. Recommendations to the authors: ESSD is not publishing research papers, the focus of the manuscript needs to be on data sets and products.

Please change your title accordingly, the focus needs to be on the main data products, e.g. Landsat derived Lake bathymetry, …

Throughout the manuscript: avoid the term 'study', introduce that you provide datasets and description of the data.
P2, L47 Deep lakes that are the focus of this study are located on the Pleistocene Sand Sea -> change this sentence that you provide a lake bathymetry data set of deep lakes that are located ….

L65 delete 'The ultimate goal of this research' – change to a sentence that includes that you provide data sets

In fact, you provide at least two data sets that you need to list in the abstract, results and conclusions: the sonar-derived lake depth data and the Landsat-derived lake bathymetry raster data. You even have produced a 3rd data set: the lake water volumes, consider to also publish the lake volume data set with Lake ID, coordinates and volume data

**Chapter 2 Data and methods**

provide a flow chart of sources, processing steps and outcomes

These are editorial requirements: Please stay consistent: provide separate chapters on sonar data acquisition and processing

and on Landsat satellite data processing – with subchapters for the Landsat satellite data processing until the final product -keeping the sonar-derived lake depth data generation separate

2.1 Field methods -> change to Lake bathymetric data

Include in this chapter all processing steps of the sonar data processing

single-beam soundings?

Move several parts of chapter 3 results to chapter 2: content on P 5 chapter 3, Lines 242 to P 6 L258, and all content on P 6 from  L268 on

As ESSD cannot publish a remote sensing research study paper this parts belong to Data and Methods

Include an estimate of the accuracy of the lake water volume data derived from the accuracy of the lake depth data.
* * *
These are editorial requirements: please provide in results a chapter with a detailed description of your published lake bathymetry data sets and lake water volume data

You can e.g. show frequency distributions, discuss shallow and deep lakes, are they evenly distributed? skewness or symmetry of lake depth data,

Show a figure of the mosaic of all bathymetry lake data with lake IDs

These are editorial requirements:

As you apply optical remote sensing for the derivation of your data set, you need to include more relevant work, specifically authors who also applied band ratio –based methods:

Please include

Clark et al. 1987. Bathymetry calculations with Landsat 4 TM imagery under a generalized ratio assumption. Appl. Opt. doi: 10.1364/AO.26.4036_1

Hodúl et al. 2018. Satellite derived photogrammetric bathymetry. ISPRS J. Photogramm. Remote Sens. doi: 10.1016/j.isprsjprs.2018.06.015.

Pacheco et al. 2015. Retrieval of nearshore bathymetry from Landsat 8 images: A tool for coastal monitoring in shallow waters. Remote Sens. Environ. doi: 10.1016/j.rse.2014.12.004

Pope et al 2016. Estimating supraglacial lake depth in West Greenland using Landsat 8 and comparison with other multispectral methods. Cryosphere. doi: 10.5194/tc-10-15-2016.

Yunus et al 2019 Improved Bathymetric Mapping of Coastal and Lake Environments Using Sentinel-2 and Landsat-8 Images. Sensors. doi: 10.3390/s19122788

Data publication: ESSD requires an optimization of the published data sets.

Please publish a new data publication with: a read me document describing the format of the data: e.g. geotiff, which projection, band variable: lake depth in meter, please include the description of the value for the land around the lake.

Editorial requirements: in the existing data publication, currently the land surrounding in at least some of the lake files seems to be noisy, consisting of 2 values: minus 1 and other values, please provide a consistent value for the land background only, it would be optimal if this would be NaN (no

data value). Please check also the land background in the mosaic if it consists of one data value only, if not please publish also an enhanced raster version of the mosaic.

---

## Author Response (AR2)

Thank you for the helpful comments and recommendations to improve this paper. Overall, we have made changes to direct focus more specifically on the datasets provided. Comments and our responses are provided in-line below (EC are editor comments; AR are author responses):

**EC: Please change your title accordingly, the focus needs to be on the main data products, e.g. Landsat derived Lake bathymetry, …**
AR: Changed to: Landsat-derived bathymetry of lakes on the Arctic Coastal Plain of Northern Alaska

**EC:Throughout the manuscript: avoid the term 'study', introduce that you provide datasets and description of the data.**
AR: We have removed the term 'study' as a descriptor of this project (however I have kept the term to refer to "study lakes" as a shorthand to specify lakes at which measurements were collected).

**P2, L47 Deep lakes that are the focus of this study are located on the Pleistocene Sand Sea -> change this sentence that you provide a lake bathymetry data set of deep lakes that are located ….**
AR: Changed to: "We collected depth measurements and mapped bathymetry at a group of deep lakes located on the Pleistocene Sand Sea…"

**EC: L65 delete 'The ultimate goal of this research' – change to a sentence that includes that you provide data sets**
AR: I have changed this sentence to: "Bathymetry measurements and associated estimates of water volume such as those provided in our datasets are important when…". This sentence better focuses on the data we provide while still conveying to a reader why they might be interested in our data.

**EC: In fact, you provide at least two data sets that you need to list in the abstract, results and conclusions: the sonar-derived lake depth data and the Landsat-derived lake bathymetry raster data. You even have produced a 3rd data set: the lake water volumes, consider to also publish the lake volume data set with Lake ID, coordinates and volume data**
AR: The 2 datasets are listed in the last sentence of the abstract and briefly described earlier in the abstract (Lines 22 – 27: "Here, we collect in situ bathymetric data to test 12 model variants for predicting Sand Sea lake depth based on analysis of Landast-8 Operational Land Imager (OLI) images. Lake depth gradients were measured at 17 lakes in mid-summer 2017 using a HumminBird 798ci HD SI Combo automatic sonar system. The field measured data points were compared to Red-Green-Blue (RGB) bands of a Landsat-8 OLI image acquired on 8 August 2016 to select and calibrate the most accurate spectral-depth model for each study lake and map bathymetry"). We have also added information to the results to focus on the data. We have also added a topic sentence to the conclusions section that focuses on the datasets: "This work provides a unique in situ depth dataset for lakes on the ACP and leverages these data alongside satellite remote sensing to map lake bathymetries and estimate volume." As the volume dataset is built into the bathymetry rasters, we feel it is unnecessary to publish the list of lake volumes as an individual dataset. Instead we provide the information in Table 4. This information is very simply derived from the bathymetry rasters.

**Chapter 2 Data and methods**
**EC: provide a flow chart of sources, processing steps and outcomes**
AR: We have added a flow chart (Figure 5).

**EC: These are editorial requirements: Please stay consistent: provide separate chapters on sonar data acquisition and processing**
**and on Landsat satellite data processing – with subchapters for the Landsat satellite data processing until the final product -keeping the sonar-derived lake depth data generation separate**
AR: As requested, we have edited the chapter names/broken up sections into 2.1 Depth data acquisition, 2.2 Depth data processing, 2.3 Landsat image selection, 2.4 Landsat image processing and analysis, 2.5 Spectral-depth point extraction, and 2.6 Model application for lake bathymetry mapping. We have also reorganized this section to better separate the different data and processing methods (e.g. moving the sentence beginning " Top-of-Atmosphere (TOA) reflectance values from the blue band (band 2; 452 - 512 nm)…" into the newly-created section 2.5.

**EC: 2.1 Field methods -> change to Lake bathymetric data**
**Include in this chapter all processing steps of the sonar data processing**
**single-beam soundings?**
AR: This subchapter name has been changed from Field methods to Depth data acquisition to ensure language is maximally specific to the subchapter content (as presumably intended by recommended revision), while also ensuring that chapter name is not confusing to readers. Changing the header here to Lake bathymetric data may cause confusion as the full bathymetry maps are not discussed until the last chapter of the Data and methods section. Instead, the last section (2.6) has been changed from Model application and volume estimation to Model application for lake bathymetry mapping. We hope this will redirect focus onto the data we provide as intended by this recommendation. In addition, as recommended above, a second section (2.2) has been created headed "Sonar data processing" to separate the data acquisition from processing steps.

**EC: Move several parts of chapter 3 results to chapter 2: content on P 5 chapter 3, Lines 242 to P 6 L258, and all content on P 6 from L268 on**
**As ESSD cannot publish a remote sensing research study paper this parts belong to Data and Methods**
AR: We have made major changes to the methods and results sections, however we believe that some discussion of the best models and their limiations is still relevant to the results and discussion sections as these were needed to create the bathymetry maps. WE have therefore left the paragraph beginning: "The best model variants… fall below this threshold" in the results section with a minore revision to the first sentence to refocus on model accuracy as a determinant of bathymetry map accuracy.

**EC: Include an estimate of the accuracy of the lake water volume data derived from the accuracy of the lake depth data.**
AR: We include this in Table 4.

**EC:These are editorial requirements: please provide in results a chapter with a detailed description of your published lake bathymetry data sets and lake water volume data**
**You can e.g. show frequency distributions, discuss shallow and deep lakes, are they evenly distributed? skewness or symmetry of lake depth data,**

AR: We have changed the focus of the results section to better reflect the importance of the data.

**EC: Show a figure of the mosaic of all bathymetry lake data with lake IDs**
AR: We have created this figure (Figure 6).

**EC: These are editorial requirements:**
**As you apply optical remote sensing for the derivation of your data set, you need to include more relevant work, specifically authors who also applied band ratio –based methods:**
**Please include (Clark et al., 1987; Hodúl et al., 2018; Pacheco et al., 2015; Pope et al., 2016; Yunus et al., 2016)**
**Clark et al. 1987. Bathymetry calculations with Landsat 4 TM imagery under a generalized ratio assumption. Appl. Opt. doi: 10.1364/AO.26.4036_1**
**Hodúl et al. 2018. Satellite derived photogrammetric bathymetry. ISPRS J. Photogramm. Remote Sens. doi: 10.1016/j.isprsjprs.2018.06.015.**
**Pacheco et al. 2015. Retrieval of nearshore bathymetry from Landsat 8 images: A tool for coastal monitoring in shallow waters. Remote Sens. Environ. doi: 10.1016/j.rse.2014.12.004**
**Pope et al 2016. Estimating supraglacial lake depth in West Greenland using Landsat 8 and comparison with other multispectral methods. Cryosphere. doi: 10.5194/tc-10-15-2016.**
**Yunus et al 2019 Improved Bathymetric Mapping of Coastal and Lake Environments Using Sentinel-2 and Landsat-8 Images. Sensors. doi: 10.3390/s19122788**
AR: We have included these references and in-text citation at the end of the Introduction.

**EC: Data publication: ESSD requires an optimization of the published data sets. Please publish a new data publication with: a read me document describing the format of the data: e.g. geotiff, which projection, band variable: lake depth in meter, please include the description of the value for the land around the lake.**
AR: I have uploaded a REAME.txt file to the Modeled Bathymetry Maps dataset summarizing the format, number of bands, pixel type, pixel depth, spatial reference information, resolution, band value, and background value, including a note describing the 2 different "background" values (i.e. the land value and the water value where negative depths were modeled). In addition, the format, spatial reference, band value description and unit, as well as other info are listed in a metadata pdf and xml provided within the publication content. The new DOI has been added to the text and references to replace those from the previous data publication.

**EC: Editorial requirements: in the existing data publication, currently the land surrounding in at least some of the lake files seems to be noisy, consisting of 2 values: minus 1 and other values, please provide a consistent value for the land background only, it would be optimal if this would be NaN (no data value). Please check also the land background in the mosaic if it consists of one data value only, if not please publish also an enhanced raster version of the mosaic.**
AR: There are two different "background" values in some of the rasters (-1 and -3.4028235e+38 [the value typically designating NoData in a 32 bit float raster]; this is now listed in the readme file in the data publication). These two values represent pixels at which negative depths were predicted by our model and land pixels, respectively. It was important to differentiate between these two pixel types, as not all "NoData" pixels were land, however including the true predicted negative depths may have led to confusion. Instead we chose to acknowledge the failing of the

models at predicting some shallow-water pixels by reclassifying negative depths to -1. This reclassification has been clarified in the text and is described in the data publication under Step 3 of the Methods description ("Pixels with negative depth values have been reclassified to a value of -1."), in the readme that was just added to the bathymetry map data publication, in the results section ("Pixels at which models predicted negative depths were reclassified to a secondary NoData value of -1 and ignored when calculating water volume (i.e., water volume was calculated for the surface area with predicted depths greater than zero [Fig. 6]).") and has been clarified in the caption for Figure 6.